# Mechanisms of Androgen Receptor Agonist- and Antagonist-Mediated Cellular Senescence in Prostate Cancer

**DOI:** 10.3390/cancers12071833

**Published:** 2020-07-08

**Authors:** Miriam Kokal, Kimia Mirzakhani, Thanakorn Pungsrinont, Aria Baniahmad

**Affiliations:** Institute of Human Genetics, Jena University Hospital, 07740 Jena, Germany; miriam.kokal@med.uni-jena.de (M.K.); kimia.mirzakhani@med.uni-jena.de (K.M.); thanakorn.pungsrinont@med.uni-jena.de (T.P.)

**Keywords:** prostate cancer, cellular senescence, bipolar androgen therapy, antiandrogen, androgen receptor antagonist, supraphysiological androgen levels, Src, PKB/Akt

## Abstract

The androgen receptor (AR) plays a leading role in the control of prostate cancer (PCa) growth. Interestingly, structurally different AR antagonists with distinct mechanisms of antagonism induce cell senescence, a mechanism that inhibits cell cycle progression, and thus seems to be a key cellular response for the treatment of PCa. Surprisingly, while physiological levels of androgens promote growth, supraphysiological androgen levels (SAL) inhibit PCa growth in an AR-dependent manner by inducing cell senescence in cancer cells. Thus, oppositional acting ligands, AR antagonists, and agonists are able to induce cellular senescence in PCa cells, as shown in cell culture model as well as ex vivo in patient tumor samples. This suggests a dual AR-signaling dependent on androgen levels that leads to the paradox of the rational to keep the AR constantly inactivated in order to treat PCa. These observations however opened the option to treat PCa patients with AR antagonists and/or with androgens at supraphysiological levels. The latter is currently used in clinical trials in so-called bipolar androgen therapy (BAT). Notably, cellular senescence is induced by AR antagonists or agonist in both androgen-dependent and castration-resistant PCa (CRPC). Pathway analysis suggests a crosstalk between AR and the non-receptor tyrosine kinase Src-Akt/PKB and the PI3K-mTOR-autophagy signaling in mediating AR-induced cellular senescence in PCa. In this review, we summarize the current knowledge of therapeutic induction and intracellular pathways of AR-mediated cellular senescence.

## 1. Introduction

### 1.1. Prostate Cancer (PCa) 

PCa is the most frequently diagnosed and the second leading lethal cancer among men in Western countries where the incidence has a strong correlation with aging [1]. Notably, with increasing age, the androgen level decreases, which seems to be timely associated with an increased risk of PCa. It was even shown that low testosterone levels are a predictive marker for PCa [2,3]. It is suggested that PCa progresses from a castration-sensitive PCa (CSPC) to the androgen deprivation therapy (ADT) resistant tumor termed as castration-resistant PCa (CRPC). Both CSPC and CRPC can metastasize leading to the lethal aggressive cancer [4]. Important to note is that the expression of androgen receptor (AR), also in CRPC and metastatic CRPC (mCRPC), is a key factor for tumor proliferation. Thus, the AR is also an important drug target for these aggressive AR positive cancers [5,6]. 

The standard PCa hormone therapy includes the inhibition of the AR-signaling by ADT and treatment with AR antagonists [7]. ADT inhibits AR-signaling by depriving the production of androgen, hence reducing activation of AR. However, PCa cells might activate intra-tumoral androgen biosynthesis [8]. Therefore, for full AR blockade, AR antagonists are used in addition to ADT. These therapeutic strategies provide initially beneficial outcomes, including reduction of the diagnostic marker prostate-specific antigen (PSA) and regression of tumor growth. However, in later therapeutic stage of most cases, primary resistance or adaptation by tumor cells occur associated with rising PSA levels and re-growth of tumor. Thus, the tumor eventually becomes therapy resistant, for which novel approaches and combinatorial treatments are required [9,10].

### 1.2. The Androgen Receptor (AR) and Adaptive Response of PCa

The AR is a nuclear hormone receptor and a ligand-controlled transcription factor belonging to the nuclear receptor superfamily. In the absence of ligand, the AR is primarily localized in the cytoplasm bound in a complex with multiple chaperones (heat shock proteins), which incapacitate AR from entering to the nucleus [11,12]. Upon androgen binding, AR forms a homodimer, changes its conformation including an interaction between the AR amino (N)-terminal FXXLF motif and the carboxyl (C)-terminal AF-2 domain containing the LXXLL motif and dissociation from the heat shock proteins leading to nuclear translocation [13,14]. Once in the nucleus, AR binds to androgen response elements in the promoter and enhancer regions of the target genes. FOXA1, HOXB13, and GATA2 are pioneering chromatin factors that help to mediate AR transcriptional activities [15]. Furthermore, many co-regulators are recruited, which act as chromatin modifiers, transcription and mediator complexes, the AR mediates transactivation or transrepression of target genes [16,17,18,19]. 

In addition to this genomic activity, also a rapid non-genomic signaling of AR is known. Within few minutes, androgen treatment leads to phosphorylation events of signal transduction factors such as MEK1/2, Src, Akt and ERK1/2 as well as other proteins like filamin A through Rac1 GTPase [20,21]. 

In CRPC despite resistance to ADT, the AR-signaling is still active likely mediated through adaptive responses that activate AR-signaling by other pathways [12,22,23,24,25]. Several underlying mechanisms exist, including but not limited to: (1) mutation of AR; (2) increased intra-tumoral androgen synthesis or (3) overexpression of AR as a result of copy number amplifications or enhanced AR protein stability; (4) changes in the expression of AR co-regulators to maintain or enhance AR mediated transactivation of target genes; (5) post-translational modifications of AR; and (6) activation of growth factors, kinases, and cytokine signaling pathways that crosstalk with AR signaling [25,26]. 

Since the CRPC growth depends on AR-signaling, targeting the AR is beneficial also in CRPC [25]. Important examples are the recently clinically approved AR antagonists Enzalutamide (Enz), Apalutamide (Apa) and Darolutamide that are used in therapy of mCRPC, mCSPC, and nmCRPC, respectively [27,28,29].

### 1.3. PCa Cell Response to Androgen Levels

Interestingly, some PCa cell lines, such as LNCaP, exhibit an opposing response towards androgen concentrations. The growth of the CSPC human LNCaP cell line is promoted by low, physiological androgen levels (LAL, with a range from 1 pM to 10 pM) whereas paradoxically supraphysiological androgen levels (SAL; with a range from 1 nM to 10 nM) inhibits PCa growth [30,31,32]. In addition, SAL treatment inhibits also tumor growth of CRPC cells in pre-clinical mouse model system [33]. This suggests a dual role of the AR as a tumor suppressor and an oncogene [34] revealing an androgen level-specific opposing tumor response. 

Therefore, besides novel AR antagonists, pharmacological SAL are used in clinical trials [35,36]. The treatment of mCRPC with ADT and intermittent cycles of ADT in combination with SAL are used in the so-called bipolar androgen therapy (BAT) [35]. 

### 1.4. Cellular Senescence in PCa

Interestingly, both AR antagonists and androgens at SAL can trigger cellular senescence in PCa in cell culture and ex vivo tumor samples [32,37], which might be one of the underlying mechanisms of AR ligand-mediated PCa growth inhibition.

In general, cellular senescence is originally referred to a stage of irreversible cell cycle arrest and has been proposed as one of the cancer suppression strategies [38,39,40]. It is considered as a stress response induced by external or internal stimuli like telomere erosion, oxidative stress or activation of certain oncogenes [41]. Senescent cells are highly metabolically active, but are arrested in the G0 phase of the cell cycle. Mostly, senescent cells acquire characteristic morphological changes like an increase in size, a tremendous re-arrangement of chromatin leading to senescence-associated heterochromatic foci (SAHF), and a distinct gene expression profile [42]. Furthermore, senescent cells can mediate paracrine effects to neighboring cells through secretion of cytokines, chemokines, growth factors, and proteases known as the senescence-associated secretory phenotype (SASP) [43,44,45,46]. 

Two known classical pathways to induce cellular senescence are the p53-p21^CIP1^ and the p16^INK4A^-pRb pathways [45]. However, activation of other CDK inhibitors, including p15^INK4B^, p19^INK4D^, and p27^KIP1^, have also been reported to mediate cellular senescence [38,47]. An activation of these pathways leads to the inhibition of cyclin-CDK complexes, thereby inducing hypophosphorylation of the retinoblastoma tumor suppressor protein (pRb) to enhance E2F-interaction and inhibition of E2F target genes that drive cell cycle progression [48]. Other senescence-inducing side-pathways also exist and may include PML, DEC1, mTOR and autophagy, or other factors interacting with the AR such as the inhibitor of growth family ING1 and ING2 [31,32,38,49,50,51]. Another mechanism which has been implicated with cellular senescence is autophagy [52,53]. It was identified as an effector mechanism of senescence. Activation of autophagy is controlled in part by the PI3K-mTOR pathway. Activation of autophagy by inhibiting mTOR induces premature cellular senescence [53]. Consistently, inhibition of autophagy delays the senescence phenotypes [52].

In PCa, the link between tumor suppressor proteins and AR-induced cellular senescence was also described. AR-induced cellular senescence associates with increased localization of PML tumor suppressor to SAHF [31]. Additionally, AR interacts with the tumor suppressors ING1 and ING2 that induce cell senescence. Both ING1 and ING2 interact with AR by acting as AR corepressors. Thus, the interaction of AR with ING family members links AR-signaling to ING-tumor suppressor signaling and cell senescence [50,51]. Importantly, the AR-p16^INK4A^-pRb tumor suppressor pathway seems to be an important pathway for AR-mediated cellular senescence in PCa [32].

## 2. AR Antagonist-Induced Cellular Senescence

Most AR antagonists prevent by competitive binding on one hand the binding of androgens to the AR. On the other hand, AR antagonists change the conformation of the AR and inhibit its activity as a transcription factor [7]. This conformational change may also inhibit non-genomic activities [54]. Also, the N/C interaction of the AR, required for maximal AR-mediated transactivation, is inhibited by most AR antagonists [37,55,56]. AR antagonists were selected for their inhibition of AR-mediated transactivation, which includes Ralaniten that targets the AR amino-terminus harboring the major transactivation domain of AR [57]. However, AR antagonists do not completely block all AR functions. So far, many AR antagonists induce a cellular senescence program in an AR-dependent manner in PCa cells (Table 1). This suggests that, although AR mediated transactivation is repressed by antagonists, it leads to the induction of cell senescence and tumor growth inhibition.

In general, AR antagonists can be structurally divided into steroidal and non-steroidal substances [65]. A clinically used steroidal AR antagonist is cyproterone acetate (CPA). Since CPA has partial AR agonistic activity and is known as a mild AR antagonist, it is meanwhile rarely used in PCa hormone therapy [66]. Rather, CPA is clinically more used as birth control pills and in the treatment of androgen-dependent conditions such as acne and excessive hair growth [67]. CPA antagonism is dependent on co-repressor binding to the AR. Thus, CPA inhibits more potently AR mediated transactivation in cells expressing higher levels of the co-repressor SMRT [66]. Presumably based on the partial agonistic activity of CPA, so far, the induction of cellular senescence by CPA has not been reported.

Interestingly, other steroidal compounds of the 20-aminosteroidal class act as complete AR antagonists [63]. These substances were designed and synthesized with a chiral tertiary C-20 amino-moiety. 20-aminosteroids require the AR ligand-binding domain (LBD) for their antagonism and compete for androgen binding to the AR. Some of these steroidal compounds, compound 11 and 18, prevent the recruitment of co-activators, which in turn leads to AR inhibition [63]. Especially one derivative, compound 18, shows a high AR specificity and potency to inhibit PCa cell growth. Interestingly, treating PCa cells with compound 18 did not result in enhanced apoptosis, but induced cellular senescence, which might in part explain the compound 18-mediated growth inhibition [63]. However, the detailed underlying mechanism of senescence induction remains unknown.

Several non-steroidal AR antagonists are widely used in clinics to treat PCa, most of them target the AR ligand-binding domain. The clinically used non-steroidal AR antagonist, Bicalutamide (Bic), sold under the brand name Casodex, is a first-generation antagonist [63,68,69,70]. It binds to the steroid-binding pocket of the AR LBD and mediates a conformational change, disrupts N/C interaction, and interferes with the formation of the coactivator-binding site [55,69,70]. The antagonist activity is mediated by an ineffective recruitment of coactivator proteins leading to a transcriptional inhibition of AR. Furthermore, it stimulates only weakly and more transiently the nuclear accumulation and binding of AR to chromatin compared to androgens [55,70,71]. Several large, prospective studies have shown that Bic monotherapy significantly improves the progression-free survival and reduces the risk of progression in early PCa [72,73]. Interestingly, studies reported that the treatment with Bic induces cellular senescence in PCa cell lines [58,59,60]. In line with this, the expression of both CDK inhibitors p16^INK4A^ and p27^KIP1^ are induced by Bic treatment [50,59].

A second-generation AR antagonist, Enz, is clinically used for mCRPC patients and sold under the brand name Xtandi. Enz is an oral AR antagonist targeting the C-terminal LBD [74,75,76]. The compound blocks the AR-androgen interaction, reduces the AR translocation into the nucleus, thus impairing the AR binding to chromatin and DNA, and inhibits receptor-mediated DNA transcription [74]. It has a 5-8-fold greater affinity to AR compared to Bic and does not increase the AR target gene activity [74,77,78]. In line with this, Enz decreases cell growth and on one hand was reported to induce apoptosis [79]. On the other hand, induction of cellular senescence in both CSPC LNCaP and the CRPC C4-2 cells upon Enz treatment was also reported [61,62]. Accordingly, the expression of p16^INK4A^ is induced by Enz in both cell lines. Interestingly, a recent study suggests that Enz significantly enhances radiation-induced cellular senescence in LNCaP cells by potentiating radiation-mediated DNA damage [80].

Another next generation AR antagonist is Apa, sold under the brand name Erleada [29]. It is approved for the treatment of mCSPC and nmCRPC [29,81]. The oral non-steroidal component binds to the LBD of AR, thereby blocking the AR translocation into the nucleus, DNA binding and AR-mediated transcription [81]. It binds to the AR with a 7–10-fold greater affinity compared to Bic [82]. Additionally, similar to Enz, Apa lacks agonistic activity. Moreover, it has been shown in CRPC murine xenografts that Apa has a greater efficacy in per unit dose and per unit steady-state plasma concentration than Enz [83]. However, a clinical study showed that AR splice variants, alterations in the AR LBD and amplifications were linked to a faster progression after Apa + ADT treatment [84]. The induction of cellular senescence by Apa has not yet been reported. 

Due to an intermediate response characterized by slowly rising PSA, and patients who do not respond at all to Enz and Apa treatment [85], additional AR antagonists are needed. Darolutamide, also known under the developmental code name ODM-201, is a second-generation non-steroidal orally active anti-androgen [86]. The mechanism of action is similar to Enz, but it does however not cross the blood-brain barrier, thus side effects such as seizures are dispensed [87]. It is a full- and high-affinity AR antagonist, which overcomes resistance to AR-targeted therapies by antagonizing both overexpressed and mutated ARs [85,88]. Darolutamide treatment results in growth inhibition and cellular senescence induction in both CSPC and CRPC PCa cell lines [62]. Similar to Enz, the induction of cellular senescence is associated with upregulated expression of the senescence regulator p16^INK4A^ in both LNCaP and C4-2 cells [62].

The benzolic methylester, atraric acid (AA), is the first described natural AR antagonists and non-steroidal compound. It was isolated from bark extracts of the African tree *Pygeum africanum,* which is sold under the trade name Tadenan© to treat prostate adenoma [37,89,90]. AA is structurally very distinct compared to Bic or Enz and a small molecule compared to steroids [91]. Nevertheless, computational analyses suggest binding of only one molecule of AA into the ligand-binding pocket of the AR [90]. For antagonism, AA binds to the LBD of AR, thereby inhibiting the N/C interaction and translocation of AR into the nucleus. AA decelerates the agonist-induced AR nuclear translocation by either retention in the nucleus or increased export of AR into the cytoplasm. Consequently, AA reduces the DNA binding of AR and thus inhibits the expression of AR target genes [37,92]. Further, AA inhibits the PCa invasiveness through the extracellular matrix [89,92]. Importantly, AA is able to induce cellular senescence in human PCa cells and ex vivo in tumor samples from patients that underwent radical prostatectomy [37]. The treatment of AA leads to an increased expression level of p16^INK4A^ and to a hypophosphorylation of the Rb protein, whereas the p53-p21^CIP1^ pathway is not significantly affected [37]. 

Another novel chemical platform that provides a leading structure for novel and specific AR antagonists includes halogen-substituted anthranilic acid esters. Specifically, the substitution with a halogen group at the benzene ring, exemplified by the compound C28, strongly inhibits the androgen-induced transactivation, chromatin, and DNA recruitment, as well as cell proliferation. Interestingly, cellular senescence is induced by these AR antagonists [64]. An inhibition of the N/C interaction was not detected, distinguishing it from other AR antagonists and suggesting that the N/C interaction of AR is not necessarily needed for AR antagonism [64] and to induce cell senescence. Another important distinction of the C28 activity is that it inhibits those AR mutants that are activated by other AR antagonists, which indicates that C28 uses a different molecular mechanism for antagonism to inhibit the AR [64,93].

Taken together, the molecular mechanisms to inactivate AR-mediated transactivation by AR antagonists are distinct among the various compounds. Nevertheless, AR antagonists induce cellular senescence despite the inhibition of AR-mediated transactivation, indicating that they do not completely block all AR functions. These observations suggest that AR antagonist-mediated cellular senescence does not process through a classical genomic AR-signaling pathway.

## 3. Supraphysiological Levels of Androgens Induce Cellular Senescence

Supraphysiological levels of androgens inhibit the growth of CRPC in vitro [32]. Clinical trials using BAT showed that the administration of supraphysiological testosterone levels (>700 ng/dL) was well tolerated in men with CRPC [94]. During BAT, the testosterone level is first elevated to supraphysiological level followed by a decline below normal testosterone level (<130 ng/dL) under ADT conditions in a 28-day treatment cycle. Studies could show that 50% of the patients had a radiographic response, whereas even 100% of the BAT-treated patients responded to second-line therapies [94]. These data suggest the ability of BAT to reverse ADT resistance. Another study supports the idea of BAT showing that the increased nuclear AR, a response to the low androgen environment, is over-stabilized after administration of SAL. This is suggested to prevent AR degradation in mitosis and inhibition of DNA replication licensing, leading eventually to PCa cell death [33]. Also, the expression of AR-responsive genes is dependent on the androgen concentration [95]. It has been shown that some genes, e.g. AR, involved in androgen synthesis, DNA synthesis, and proliferation, are repressed upon high androgen levels. Therefore, Isaacs et al. suggested that BAT might prevent adaptive changes in AR expression leading to a prolonged therapy response of patients [95]. However, some patients developed a resistance also towards this treatment option, wherefore further improvements are required [96].

Accordingly, the treatment of PCa cells with the natural androgen dihydrotestosterone (DHT) or the synthetic androgen methyltrienolone (R1881) leads to a concentration-dependent proliferation arrest [30,31,32,97]. In line with this, LAL stimulated proliferation, whereas SAL treatment of either DHT or R1881 induced cellular senescence and inhibited growth in PCa cells [32] (Table 1). Consistent with SAL-induced cellular senescence, an increased percentage of cells in G1/G0 phase of cell cycle is observed by SAL treatment. Mechanistically, SAL treatment resulted in an increased level of the tumor suppressor p16^INK4A^ as well as hypophosphorylation of pRb, downregulation of E2F and cyclin D1 [32]. Knock-down of p16^INK4A^ suggests that SAL-induced cellular senescence is mediated through the p16^INK4A^-pRb pathway. Moreover, cellular senescence was induced by SAL treatment in PCa samples ex vivo obtained from patients with prostatectomy [32]. 

Notably, AR-negative PCa cells do not show SAL-dependent change of cell proliferation or induction of cellular senescence. Using an inducible system in AR-negative cells, SAL induces cellular senescence, revealing an AR-dependent effect [31,32]. Thus, AR-independent effects by SAL at cell membranes are likely excluded.

## 4. Interplay between AR-Signaling and other Cellular Signaling Pathways in Senescent PCa

The classical AR-signaling, with AR as a ligand-controlled transcription factor, is known as a genomic pathway, which is suggested to occur over several hours [98]. However, there are several cellular responses to AR-signaling that do not follow this classical genomic pathway. The non-genomic androgen signaling was observed within few minutes after androgen treatment and resulted in phosphorylation of MEK1/2, Akt and Erk1/2 [20]. Within a few hours, these phosphorylation events disappeared, leading to the term rapid signaling.

It is meanwhile known that by androgen treatment not all AR molecules translocate into the nucleus. Rather a minor fraction remains in the cytoplasm and interacts with lipid rafts containing Akt [99]. Also, other cytosolic and membrane-associated factors such as MAPK, Src, Akt/PI3K interact with AR [99,100,101,102,103,104], leading to phosphorylation events and the activation of intracellular kinase cascades. There are cross-phosphorylation events such as between Akt and AR. AR leads to an increase of Akt-phosphorylation and Akt on the other hand phosphorylates AR.

The activation of Src pathways is often observed in PCa [101,105]. Src is the prototypical member of the largest family of non-receptor tyrosine kinases (TKs) including nine members. TKs include mostly membrane associated receptors and non-receptor TKs. Src is known as a junction of several pathways, including AR, TGFβ, Bcl-2, Akt/PTEN or MAPK, and ERK [101]. AR interacts with Src by binding to a polyproline sequence between residues 371 and 381 of the AR N-terminal domain to the Src homology domain 3 [102]. This binding activates the Src kinase domain [106,107]. On one hand, a rapid association of AR with Src leads to the enhanced cell proliferation through activation of the MAPK/ERK cascade. On the other hand, this interaction decreases cell proliferation and increases cellular senescence in PCa cell lines in the presence of SAL [32]. Interestingly, the total amount of Src seems to be unchanged after SAL treatment whereas the ratio between Src and phospho-Src is changed towards an enhanced Src phosphorylation level. In LNCaP cells, the Src inhibitor PP2 reduces the androgen-mediated cellular senescence after SAL treatment, while upon LAL no detectable influence on cell senescence was observed. However, blocking MEK1/2-kinases, a downstream target of the Src-kinase, with the U0126 inhibitor did not affect androgen-induced cellular senescence. Similarly, phosphorylation level of ERK1/2 reveals no obvious change by SAL. Furthermore, using specific inhibitors for other Src downstream factors such as p38 and STAT3 exhibited similar results and did not reduce the level of cellular senescence [32]. This indicates that Src mediates androgen-induced cellular senescence through other downstream pathways (Figure 1). Interestingly, treating the PCa cells with Src inhibitor PP2 also reduces the AR antagonist-mediated cellular senescence after AA treatment [37]. This indicates that the Src pathway is also involved in AR antagonist-induced cellular senescence (Figure 2).

Therefore, a link between AR-induced cellular senescence and the PI3K/Akt/mTOR has also been suggested. Phosphoinositide 3-kinases (PI3Ks), also called phosphatidylinositol 3-kinases, are aberrantly activated by various genetic and epigenetic alterations in PCa [108,109]. The PI3K family is divided into four different classes: Class I, Class II, Class III, and Class IV, which play a crucial role in different cellular functions. Many of these functions relate to the ability of class I PI3Ks to activate protein kinase B (PKB/Akt) as in the PI3K/Akt/mTOR pathway. Akt is activated by phosphorylation [110] leading to phosphorylation of a number of important effectors, which regulate a variety of processes that coordinate cell survival [111]. It has been shown that the PI3K pathway is commonly deregulated in PCa and regulates the tumor development and progression through different pathways, including loss of PTEN, which promote cell survival and proliferation, metastasis, and angiogenesis in a cell-intrinsic and -extrinsic manner [108,109]. Unfortunately, the PI3K pathway drives resistance to different anticancer therapies [112,113]. In PCa, the progression to CRPC after ADT is closely associated with the PI3K complex interaction with the AR pathway [114,115,116]. Thus, inhibitors of this pathway could be a beneficial therapy for patients with CRPC. So far as monotherapy, these inhibitors failed to achieve clinically significant responses in PCa [117,118]. Nevertheless, studies showed that the PI3K inhibitor, 3-MA, reduces the level of androgen induced cellular senescence in LNCaP cells [32] suggesting the involvement of PI3K in androgen-induced cell senescence (Figure 1). 

Also, the Akt family has been shown to drive PCa formation in vivo [119]. A large number of CRPC patients have activation of the PI3K/AKT/mTOR pathway mostly by mutations of PTEN [120]. Akt interacts with AR and phosphorylates AR in its N-terminus. The phosphorylation of AR by Akt at Serine 213 is associated with reduced survival of CRPC patients and suggests that the activation of Akt is an important driver for castration resistance [121]. The underlying molecular mechanism by which AR antagonists induce cell senescence through modulating the AKT signaling is unclear.

Interestingly, Akt substrates play different roles in the AR pathway. Studies showed that the Akt target FOXO3a increases AR expression via direct binding to the AR promoter [122], while FOXO1, has been shown to recruit histone deacetylase HDAC3 and may reduce the AR transactivation [123]. In addition, the residues S213 and S791 of AR, which are phosphorylated by Akt, activate the transactivation [124]. It has been shown that Akt activation induces proliferation and survival of mammalian cells [125]. Some studies reported an increased level of Akt (S473) in high Gleason grade PCa [126]. In contrast, studies showed that upon SAL condition, the protein level of p-Akt (S473) is increased in LNCaP cells but not after LAL treatment [32]. Interestingly, several studies reveal that an Akt inhibitor induces the level of cellular senescence in different cancer cell types such as PCa and papillary thyroid carcinoma (PTC) [127,128]. However, using a specific Akt inhibitor (Akti, 1L6-Hydroxymethyl-chiro-inositol-2-(R)-2-O-methyl-3-O-octadecyl-sn-glycerocarbonate) reveals a strong reduction of the SAL-mediated cell senescence [32]. Interestingly, Akti treatment also reduces the levels of AA-mediated cell senescence [37]. Since both Src and Akt are known interaction partners of AR and both phosphorylate AR [129], these data suggest that the Src-Akt pathway is involved in both SAL- and the antagonist AA-mediated cellular senescence in PCa cells (Figure 1 and Figure 2). Interestingly, in different sarcoma cell lines, the treatment with Akti disrupts the translocation of Akt to the plasma membrane [130]. Thereby, the activation by membrane-bound phosphatidylinositol-dependent kinases is inhibited [131]. Based on these data, one can speculate for SAL-induced cellular senescence that either the AR-Akt complex at the membrane and/or the activation of phosphatidylinositol-dependent kinases is required. 

For prostate tumorigenesis, it seems that the downstream effector of Akt signaling, the serine/threonine protein kinase mammalian target of rapamycin (mTOR), is essential [132]. mTOR forms the catalytic subunit of two distinct complexes: mTOR complex 1 (mTORC1) and 2 (mTORC2) [115]. It has been shown in non-tumoral vas deferens epithelial cells (VDEC) or PC3 cells, when stably expressing AR, that phospho-Akt at serine 473 is increased in less than an hour upon androgen stimulation [133]. Additionally, the role of the mTORC1 in driving SASP has been widely established [134]. Furthermore, mTORC1 can be targeted to eliminate senescent cells [134]. In senescent cells, the expression of autophagy and lysosome proteins is dramatically enhanced. This effect may relate to mTORC1 [135]. Activated mTORC1 inhibits the catabolic process of autophagy to support cell growth and metabolism. mTORC1 is switched off and autophagic activity increases in courses of limiting nutrient access, and delivering cytoplasmic material to the lysosomes for degradation. Salvation free amino acids, lipids and carbohydrates from the lysosome are supporting cell survival [136]. Metabolism and survival of senescent cells excessed endure rewiring of these pro-growth [134,137]. Therefore, senescence results in an inimitable balance between mTORC1 and autophagy. Interestingly, SAL influenced autophagy activity [32]. Rapamycin an inhibitor of mTOR co-treated with SAL resulted in reduction of cell senescence [32] indicating that autophagic activity is associated with SAL-induced cell senescence.

Taken together, these findings suggest a non-genomic, but not necessarily rapid signaling, of AR-mediated pathways to induce cellular senescence. In line with the non-genomic action, so far many known AR antagonists reduce nuclear localization, which could be mediated by an increase of nuclear export and/or enhance nuclear mobility as a characteristic of reduced chromatin binding. Since these AR antagonists induce cell senescence, it may be hypothesized that cytosolic non-DNA-bound AR mediates cellular senescence, which eventually leads to changes in the transcriptome (Figure 1 and Figure 2).

## 5. Targeting AR Ligand-Induced Cellular Senescent PCa Cells with Senolytic Compounds

The benefit of senescence induction is controversial due to emerging reports on tumor promoter effects of SASP [43,44,45]. Hence, targeting specifically senescent tumor cells and their elimination by senolytic compounds could be an interesting approach in PCa therapy. Senolytic compounds are molecular compounds that induce cell death in senescent cells by targeting an activated pro-survival/anti-apoptotic pathway [138]. 

Interestingly, different senescence inducers can lead to distinct activation/upregulation of pro-survival/anti-apoptotic pathways [61,138,139]. This has been observed as well in case of AR ligand-induced cellular senescence in PCa. Our recently published data indicate that the pro-survival Akt-S6 pathway is more activated by AR agonist-induced cellular senescence compared to antagonist, although both types of ligand induce cell senescence by AR [61]. In line with this, SAL treatment led to increased phosphorylation of both Akt and S6, whereas Enz treatment induced phosphorylation of Akt but not S6. This different regulation of the Akt-S6 pathway consequently led to distinct outcomes for the senolytic-treated senescent PCa cells. 

Inhibition of Akt by highly selective allosteric Akt inhibitor MK2206 sensitized AR antagonist-treated cells to apoptosis, while SAL-treated cells were resistant [61]. Importantly, inhibition of Akt by MK2206 reveals that phosphorylation of S6, a downstream target of Akt, might be regulated in part by AR ligands, independent of Akt phosphorylation. The phospho-S6 levels remained high although Akt was totally inhibited under SAL-treated condition, which was not observed in Enz-treated cells. Thus, AR agonist or antagonist affects differently downstream Akt signaling. 

Interestingly, although SAL-mediated cellular senescent PCa cells seem to be resistant for senolytic agent MK2206, they are sensitive to another senolytic agent Ganetespib, a HSP90 inhibitor [61]. The data show that phosphorylation of S6 was also inhibited by HSP90 inhibitor under SAL-treated condition. Notably, phospho-S6 has been suggested as a critical pro-survival factor [140,141,142]. Hence the levels of phospho-S6 and distinct regulation of this factor by AR ligands might be an underlying mechanism for sensitivity against specific senolytic compounds.

## 6. Conclusions

In conclusion, both androgens at supraphysiological levels (Figure 1) and AR antagonists (Figure 2) induce cellular senescence in PCa. This important AR pathway is mediated by membrane and cytosolic transduction factors including PI3K, Src family, Akt and mTOR. AR was shown previously to interact in a non-genomic and rapid signaling with Src and Akt. Analyzing AR ligand-induced cell senescence, the activation of these factors was however also observed after many days of AR ligand treatment. Therefore, it is suggested that the AR interacts with these factors at the non-genomic level, also in a long-term manner, which eventually changes the transcriptome landscape. Interestingly, despite both ligands inducing cancer cell senescence, AR agonist and antagonist seem to induce a distinct pro-survival pathway. Therefore, targeting senescent PCa cells, the specific pro-survival pathway should be known in order to use a particular senolytic compound.

## Figures and Tables

**Figure 1 cancers-12-01833-f001:**
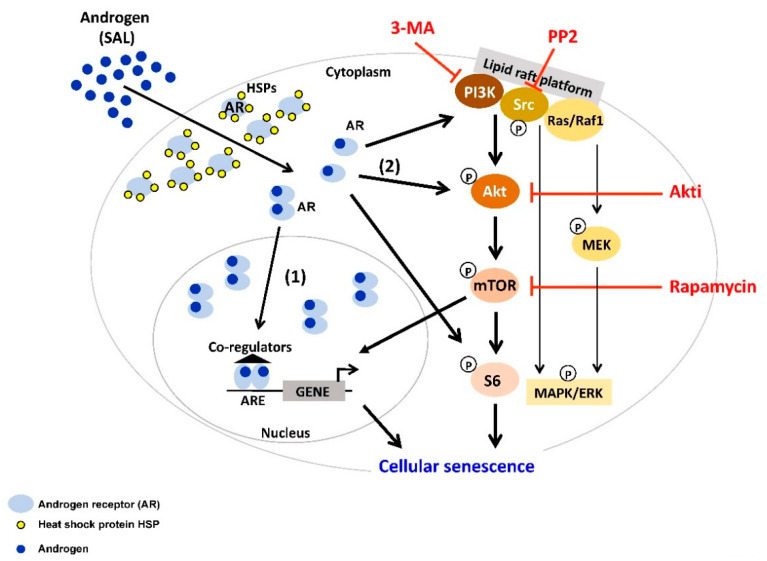
Androgen at supraphysiological level (SAL) induces cellular senescence through the PI3K/Src/Akt/mTOR pathway. In the cytoplasm, androgen receptor (AR) forms a complex with heat shock proteins (HSPs). Activation of the AR is triggered by binding of androgens (AR agonist) to the receptor. Androgen-activated AR dissociates from HSPs, and subsequently mediates AR signaling. (1) The majority of activated AR forms homodimer and translocates into the nucleus, where it binds to androgen response element (ARE) at the promoter of the target genes and mediates transcription. (2) A minor fraction of activated AR remains in the cytoplasm, interacts with and functions through several signaling molecules, e.g., Src, PI3K/Akt, S6, Ras/Raf1, MAPK/ERK. Interestingly, co-treatment of SAL with inhibitors of PI3K (3-MA), Src family members (PP2), Akt (Akti), or mTOR (Rapamycin) inhibits senescence induction. This indicates that Src/PI3K/Akt/mTOR pathway is involved in SAL-mediated cellular senescence in PCa cells.

**Figure 2 cancers-12-01833-f002:**
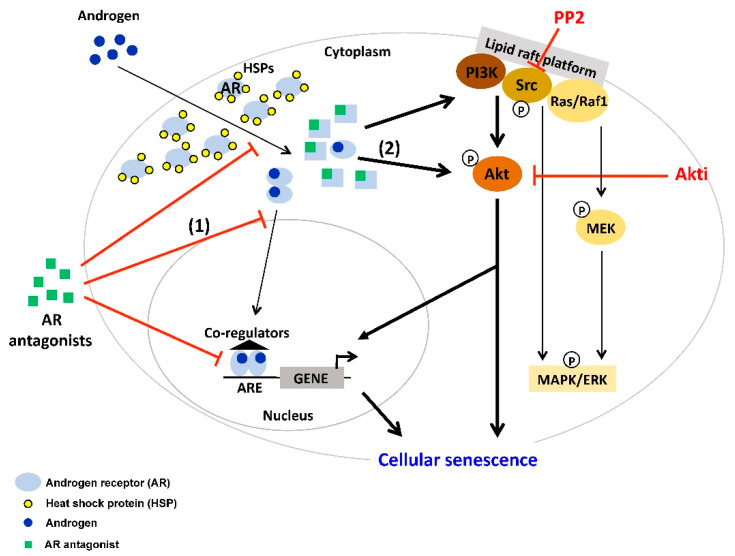
AR antagonist induces cellular senescence in PCa through the Src/Akt pathway. Binding of AR antagonist to AR leads to conformational change of AR. AR antagonists compete with androgens for binding to AR. (1) Moreover, some AR antagonists reduce AR nuclear translocation, chromatin association and DNA binding. (2) As a result, the majority of AR resides in the cytoplasm and may interact with cytoplasmic factors to mediate cellular senescence. Co-treatment of the AR antagonist atraric acid with the inhibitors of Src (PP2) or Akt (Akti) reduce cell senescence suggesting that Src/Akt pathway is involved in AR antagonist-induced cellular senescence in PCa cells.

**Table 1 cancers-12-01833-t001:** List of AR ligands, which induce cellular senescence.

AR Ligands	Detection of Cellular Senescence and Molecular Pathways	Cell Lines and PCa Tissues	References
**AR antagonists**			
Bicalutamide	SA-β-gal, p16^INK4A^, p27^KIP1^	LNCaP, PC3 AR, CWR22PC	[50,58,59,60]
Enzalutamide	SA-β-gal, p16^INK4A^	LNCaP, C4-2	[61,62]
Darolutamide	SA-β-gal, p16^INK4A^	LNCaP, C4-2	[62]
Atraric acid	SA-β-gal, p16^INK4A^, pRb, Src, Akt	LNCaP	[37]
Novel 20-aminosteroid (Compound 18)	SA-β-gal	LNCaP	[63]
Halogen-substituted anthranilic acid esters	SA-β-gal	LNCaP	[64]
**AR agonists**			
Dihydrotestosterone	SA-β-gal, SAHF, p14^ARF^, p16^INK4A^, p21^CIP1^, Cyclin D1, pRb, p63, mTOR, ROS, PML	PC3 AR, LNCaP, C4-2, RWPE AR, PCa tissue ex vivo	[31,32,50]
Methyltrienolone	SA-β-gal, SAHF, p14^ARF^, p16^INK4A^, p21^CIP1^, p27^KIP1^, Cyclin D1, E2F1, pRb, Src, Akt	PC3 AR, LNCaP, C4-2, PCa tissue ex vivo	[32,50]

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
