# Peer review of "Mechanisms of Androgen Receptor Agonist- and Antagonist-Mediated Cellular Senescence in Prostate Cancer"

_cancers, 2020, doi:10.3390/cancers12071833_

Round 1

Reviewer 1 Report

This is a good overview of the topic. It should be published, after addressing a few issues.

Comments:

1) On line 53, instead of "develops", use "progresses"?

2) On line 66, "re-raising" should be "rising"

3) In discussion about AR antagonists, authors mention enzalutamide and darolutamide, but do not mention the third approved AR antagonist, apalutamide.  It may be appropriate to discuss apalutamide around line 92-93 and after line 202, along with references.

Author Response

We would like to thank the reviewers very much for their fruitful comments.

All changes are highlighted in red.

We have dealt with the critical parts as follows:

Reviewer 1:

Thank you very much for the positive evaluation.

1) On line 53, instead of "develops", use "progresses"?

We have changed accordingly.

2) On line 66, "re-raising" should be "rising"

We have changed accordingly.

3) In discussion about AR antagonists, authors mention enzalutamide and darolutamide, but do not mention the third approved AR antagonist, apalutamide.  It may be appropriate to discuss apalutamide around line 92-93 and after line 202, along with references.

We have now mentioned Apalutamide in line 96 and added an entire paragraph with Apalutamide and completed the currently used AR antagonists. See page 5

Reviewer 2 Report

The manuscript by Drs. Kokal, Mirazakhani et al, is an excellent and timely review of the mechansim by which the Androgen Receptor (AR) agonists and antagonists work on prostate cancer. Importantly, the authors discuss the senesence inducing effect of AR agonists on Castration Resistant prostate cancer (CRPC) where supraphysiological levels of AR agonist seem to halt the growth of PCa by inducing senescence associated phenotype but not autophagy or apoptosis. The authors have done an outstanding job of a systematic review of previous and current work on both the second generation Androgen antagonists such as bicalutamide and then the third generation Androgen antagonists, enzalutamide and daralutamide and some unrelated non-steroidal inhibitors of AR function. The review however, does not dwell much into the descent of cells into senesence which at present is not accepted as the defacto mechanism much less has known mechanism. The authors do not have to tweak the manuscript much to add a little more about the mechanism by which AR antagonists in CRPC or in mCRSC make PCa descent into senescence. Regardless, the article is a good and timely review on this important topic of second line of therapy for prostate cancer. 

Author Response

We would like to thank the reviewers very much for their fruitful comments.

All changes are highlighted in red.

We have dealt with the critical parts as follows:

Reviewer 2:

Thank you very much for rating our manuscript as an excellent and timely review.

1) The authors do not have to tweak the manuscript much to add a little more about the mechanism by which AR antagonists in CRPC or in mCRSC make PCa descent into senescence.

We have now added on page 9 and 10 the link of activated Akt signaling with CRPC and some molecular possibilities of senescence induction by the AR-Akt-signaling in a separate paragraph on page 10.

Reviewer 3 Report

In this manuscript, the authors describe an important action of androgens in prostate cancer, the reduction of cell proliferation by the activation of cell senescence. The key factor of this review is the discussion about the action of supraphysiological androgen concentration because the hormone acts in a way completely opposite than usual. Probably this point of view should be reinforced and, in the same paragraph, the authors should discuss about transcriptional and non-transcriptional pathways used by androgen (SAL) to induce senescence.

I have some comments:

In this review the authors often talk about supra-physiological and physiological or low and high androgen concentration but they should specify these concentrations or the ranges in which they are included.

Lane 67: LSD1 is another demethylase that help AR to mediate its transcriptional activities (for example doi: 10.1186/s12935-018-0568-1 and many others)..

Lane 80: Considering that the previous sentence starts with "In addition" could be better substitute one of the two.

Lane 82: other proteins are Filamin A or the Rac1 gtpase (as in DOI: 10.3389/fendo.2014.00225 or in doi: 10.1371/journal.pone.0076899 or in DOI: 10.1038/cddis.2014.497 ).

Lanes 140-141: Probably, when an inhibitor modify the AR conformation, alters not only the transcriptional but also the non-transcriptional activity. This is in general. After you can specify that some inhibitors work without completely inhibit AR actions.

Lane 141: please indicate what is N/C. A

Lanes 236-238: I don't understand this sentence. Do the authors imply that after the 28 days , during which the androgen is administered and removed (BAT therapy), the level of androgen is first high and after normal? What about the therapy efficiency (growth of cancer or other) after BAT therapy?

Author Response

We would like to thank the reviewers very much for their fruitful comments.

All changes are highlighted in red.

We have dealt with the critical parts as follows:

Reviewer 3:

We thank for your fruitful comments.

1) In this review the authors often talk about supra-physiological and physiological or low and high androgen concentration but they should specify these concentrations or the ranges in which they are included.

We have now included concentrations about low and supraphysiological androgen levels both in cell culture experiments and for patients undergoing bipolar androgen therapy. See page 3 and page 6.

2) Lane 67: LSD1 is another demethylase that help AR to mediate its transcriptional activities (for example doi: 10.1186/s12935-018-0568-1 and many others).

Yes, indeed there are many AR interacting factors. To cite each of them might distract. Therefore, we referred to two comprehensive reviews and LSD1 (doi: 10.1186/s12935-018-0568-1). See page 2.

3) Lane 80: Considering that the previous sentence starts with "In addition" could be better substitute one of the two.

We have changed accordingly

4) Lane 82: other proteins are Filamin A or the Rac1 gtpase (as in (as in DOI: 10.3389/fendo.2014.00225 or in doi: 10.1371/journal.pone.0076899 or in DOI: 10.1038/cddis.2014.497).

Thank you very much. We added now these factors.

5) Lanes 140-141: Probably, when an inhibitor modify the AR conformation, alters not only the transcriptional but also the non-transcriptional activity. This is in general. After you can specify that some inhibitors work without completely inhibit AR actions.

Indeed, thank you for the comment. We added this point. See line 150.

6) Lane 141: please indicate what is N/C.

We have now explained the N/C-terminal interaction and provided the corresponding interaction motifs. See page 2.

7) Lanes 236-238: I don't understand this sentence. Do the authors imply that after the 28 days, during which the androgen is administered and removed (BAT therapy), the level of androgen is first high and after normal? What about the therapy efficiency (growth of cancer or other) after BAT therapy?

We added additional information on BAT and supraphysiological androgen levels. Patients are continuously treated with ADT and obtain cycles of SAL. See page 6, lines 256-261.

Round 2

Reviewer 3 Report

The authors completely answered to all my comments. To my humble opinion, in this form the manuscript is suitable for publication on Cancers. 

Lane 374: Please modify Akti in Akt.